# Antibiotic Resistance of *Bacillus cereus* in Plant Foods and Edible Wild Mushrooms in a Province

**DOI:** 10.3390/microorganisms11122948

**Published:** 2023-12-09

**Authors:** Xiaoyan Cha, Yingting Lin, Charles Brennan, Jianxin Cao, Ying Shang

**Affiliations:** 1Faculty of Food Science and Engineering, Kunming University of Science and Technology, Kunming 650500, China; 20212114039@stu.kust.edu.cn (X.C.); 20212114038@stu.kust.edu.cn (Y.L.); charles.brennan@rmit.edu.au (C.B.); 2School of Science, Royal Melbourne Institute of Technology University, Melbourne 3000, Australia

**Keywords:** *Bacillus cereus*, antibiotics, antibiotic resistant, phenotype, genotype

## Abstract

*Bacillus cereus* is a common pathogen causing foodborne diseases, secreting and producing a large number of toxins that can cause a variety of diseases and pose many threats to human health. In this study, 73 strains of *Bacillus cereus* were isolated and identified from six types of foods from seven different cities in a province, and the antibiotic-resistant phenotype was detected by using the Bauer–Kirby method. Results showed that the 73 isolates were completely sensitive to gentamicin and 100% resistant to chloramphenicol, in addition to which all strains showed varying degrees of resistance to 13 other common antibiotics, and a large number of strains resistant to multiple antibiotics were found. A bioinformatic analysis of the expression of resistance genes in *Bacillus cereus* showed three classes of antibiotic-resistant genes, which were three of the six classes of antibiotics identified according to the resistance phenotype. The presence of other classes of antibiotic-resistant genes was identified from genome-wide information. Antibiotic-resistant phenotypes were analyzed for correlations with genotype, and remarkable differences were found among the phenotypes. The spread of antibiotic-resistant strains is a serious public health problem that requires the long-term monitoring of antimicrobial resistance in *Bacillus cereus*, and the present study provides important information for monitoring antibiotic resistance in bacteria from different types of food.

## 1. Introduction

*Bacillus cereus* is a spore-forming Gram-positive bacillus [1], a conditional pathogen that can cause a wide range of diseases [2]. Bacteria are ubiquitous in the natural environment [3], and can be isolated from soil, sediments, water, and different types of foods [4,5]. The presence of *B. cereus* has been reported in foods such as infant formula, milk [6,7], dairy products [8,9], fruits, vegetables, preserves [10,11], and spices [12]. This bacterium has become a common foodborne pathogen worldwide [13], causing many different degrees of food poisoning [14].

The discovery of antibiotics led to the killing or inhibition of pathogenic microorganisms in the body. However, with the widespread use of antibiotics, many strains of bacteria have been found to develop resistance to antibiotics, a phenomenon known as antibiotic resistance. Studies have shown that this antibacterial phenomenon existed long before antibiotic resistance was discovered [15]. The emergence and spread of antibiotic resistance is a worldwide problem that poses a serious challenge to researchers around the world. The World Health Organization has ranked antibiotic resistance as a health challenge of the 21st century [16]. According to a recent report, by 2050, 10 million people will die globally each year from antibiotic resistance if no intervention is developed [17]. With the emergence of a large number of bacterial pathogens resistant to multiple antibiotics, multi-antibiotic-resistant bacteria have been considered a global emergency [18], posing a serious threat to human health [19]. Some studies have shown that it takes about 2 years for bacteria to go from sensitivity to resistance to antibiotics, while the development of a novel antibiotic takes about 10 years [20]. The speed of the transition from sensitivity to resistance is faster than the development cycle of antibiotics, so accelerating the development of novel antibiotics and slowing down the rate of bacterial resistance are crucial [21].

Antimicrobial resistance genes are the material basis for the formation and transmission of bacterial resistance, which can be transmitted to other bacteria through bacterial transformation, transduction, and conjugation, and can even be transmitted among bacteria of different species [22]. *B. cereus* usually produces *β*-lactamases; thus, it is remarkably resistant to *β*-lactam [23]. On the contrary, *B. cereus* strains are sensitive to aminoglycosides, clindamycin, chloramphenicol, erythromycin, and vancomycin. Some strains are resistant to tetracycline, streptomycin, ciprofloxacin, cloxacillin erythromycin, and rifampicin [24].

The consumption of fresh vegetables is one of the main routes through which antibiotic resistant bacteria and antibiotic resistant genes are transferred from the natural environment to humans, posing potential health risks [25]. Thus, the consumption of foods contaminated with *B. cereus* can lead to antibiotic-resistant infections in humans. The detection rate of *B. cereus* in Chinese vegetables is as high as 50% [26]; *B. cereus* isolated from raw vegetables is sensitive to imipenem, vancomycin, gentamicin, erythromycin, ciprofloxacin, and chloramphenicol [27]. *B. cereus* isolated from garlic chives shows moderate resistance to cefotaxime, rifampicin, clindamycin, erythromycin, and tetracycline [28]. The antibiotic resistance profile of *B. cereus* strains isolated from Mexican chili powder shows high resistance to *β*-lactams, trimethoprim/sulfamethoxazole, tetracycline, erythromycin, clindamycin, and chloramphenicol [29]. The multi-drug resistant phenotype of *B. cereus* in rice samples was observed [3]. *B. cereus* isolated from preserved bean curd was susceptible to 11 antibiotics: penicillin, gentamicin, tetracycline, erythromycin, chloramphenicol, ciprofloxacin, benzathine, clindamycin, rifampicin, amikacin, and cotrimoxazole [21]. Briefly, 94.2% of *B. cereus* isolated from frozen foods were identified as multi-antibiotic resistant strains [30]. In ensuring human health and food safety, detecting antibiotic resistance in food products isolated from *B. cereus* is important.

In recent years, foodborne diseases caused by *B. cereus* have been increasing annually, bringing serious impacts on people’s health and social development [31,32,33]. The dietary structure of the area involved in this study is rich and varied, particularly in terms of foods such as wild mushrooms, soybean products, and pickled vegetables being popular among local people. However, these foods are highly susceptible to contamination by *B. cereus* during production, processing, and transportation, leading to food poisoning, which is a serious threat to people’s health. Hence, this study aimed to detect antibiotic-resistant phenotypes and analyze resistant genes in 73 strains of *B. cereus* isolated from five types of plant foods and edible wild mushrooms.

## 2. Materials and Methods

### 2.1. Sample Collection

The 273 samples used in this study were randomly collected from seven different areas of a province (wild mushroom, *n* = 32; soybean products, *n* = 76; fresh vegetables, *n* = 80; preserved vegetables, *n* = 47; cereals, *n* = 4; frozen food cereals, *n* = 34). Samples were placed in sterile sampling bags immediately after purchase, placed in a cryogenic sampling box, and brought back to the laboratory within 24 h.

### 2.2. Isolation of B. cereus

Ten grams of a sample was weighed using a sterile homogenizer cup with 100 mL of physiological saline and homogenized at 8000–10,000 r/min for 1–2 min using a rotary blade homogenizer. In accordance with the national standard (GB 4789.14-2014) [34], a homogeneous solution was diluted at 1:100, and the mixture was cultured at 36 °C for 18–24 h. The isolation and purification of *B. cereus* were carried out using a *B. cereus* chromogenic medium (Chromagar, Central Bio-Engineering Co., Ltd., Shanghai, China) for isolation, and single colonies were collected via plate delineation and purely cultured at 36 °C for 18–24 h. After purification, single colonies were collected and cultured in LB broth (HB0128, Hope Bio-Technology Co., Ltd., Qingdao, China) and incubated at 36 °C for 18–24 h. After enrichment, the strains were preserved with 30% glycerol and placed into a −80 °C ultra-low-temperature refrigerator for long-term preservation.

### 2.3. The 16S rRNA Gene Identification of Isolates and Whole Gene Sequencing

Ezup Columnar Bacterial Genomic DNA Extraction Kit (B518255-0100) was pur chased from Shanghai Sangon Biotechnology Co., Ltd. (Shanghai, China) and used to extract DNA. Briefly, 16S rRNA was used for strain identification. The primers for 16S rRNA gene amplification were 27F primer (5′-AGAGTTTGATCCTGGCTCAG-3′) and 1492R primer (5′-RTACGGCTACCTTGTTACGACTT-3′) [35]. The 25 μL reaction system was used for PCR reactions in a thermal cycler, and the reaction system contained 2.5 μL of 10× buffer, 2 µL of a dNTP mixture (2.5 mM), 0.2 μL of rTaq DNA polymerase (5 U/µL; TaKaRa, Biotechnology, Dalian, China), 1 µL of primer F (10 µM), 1 µL of primer R (10 µM), 2 µL of DNA template, and 16.3 µL of ultrapure water. The amplification conditions for the PCR program were as follows: initial denaturation at 94 °C for 2 min; 30 cycles of denaturation at 94 °C for 30 s, primer annealing at 54 °C for 30 s, and primer extension for 1 min at 72 °C; and a final extension at 72 °C for 10 min. The PCR amplification products were quality controlled using 2% agarose gel (containing 0.1 μL/mL of TS-GelRed). Agarose was purchased from Sigma-Aldrich Chemical Co., Ltd. (Shanghai, China). In addition, the PCR amplification products were subjected to Sanger sequencing to obtain the 16S rRNA sequences of the samples, and PCR amplification and sequencing were performed by Sangon Biotech (Shanghai) Co., Ltd. The 16S rRNA sequences of the sequenced samples were uploaded to the NCBI database, and sequence comparison was performed using the BLAST tool. Moreover, strains with a homology rate greater than 99% were considered to be of the same species [36]. Finally, we performed whole gene sequencing on all isolated strains [37,38,39,40,41].

### 2.4. Antibiotic Resistance Testing of Isolate

The antimicrobial susceptibility of 73 strains of *B. cereus* was evaluated using the Kirby–Bauer disk diffusion method in accordance with performance standards for antimicrobial susceptibility testing from the Clinical and Laboratory Standards Institute (CLSI) [42] for *Staphylococcus aureus* [38]. *B. cereus* isolates in LB liquid medium were incubated at 36 °C for 24 h. The concentration of the bacterial suspension was adjusted to 0.5 McFarland (G60346) using 0.9% saline in standard tubes, which was purchased from Bio-Kont Co., Ltd. (Wenzhou, China), and the resulting suspension was used for subsequent experiments. Fifty microliters of the diluted suspension was aspirated and added to a solid (MH) medium of the antibiotic sensitivity test, and tweezers were used to hold the antibiotic-sensitized tablets on the surface of the plate. Each plate could be affixed with three tablets at an angle of 120°, and the distance between each tablet was not less than 24 mm. The plates were inverted and incubated for 24 h in a 36 °C incubator. The antibiotic disks used were 30 µg of ceftazidime (CAZ), 10 µg of cephalothin (CEP), 10 µg of ampicillin (AMP), 10 µg of amoxicillin (AMX), 30 µg of cefotaxime (CTX), 10 µg of ticarcillin–clavulanic acid (TCC), 30 µg of chloramphenicol (CHL), 30 µg of imipenem (IPM), 10 µg of streptomycin (STR), 30 µg of kanamycin (KAN), 10 µg of gentamicin (GEN), 30 µg of nalidixic acid (NAL), 5 µg of ciprofloxacin (CIP), 25 µg of trimethoprim/sulfamethoxazole (SXT), and 250 µg of sulfafurazole (SOX). The antibiotics used in this study were purchased from Oxoid Ltd. (Basingstoke, Hants, UK). The allergy test solid (MH) medium was purchased from Hangzhou Best Biotechnology Co., Ltd. (Hangzhou, China). The diameter and length of the inhibition zone of each *B. cereus* isolate were measured with a vernier caliper to explain antibiotic sensitivity [43], and the results were interpreted following the guidelines provided by the CLSI document M100, 31st edition [42,44]. To evaluate the efficacy of the antimicrobial susceptibility testing process [45], *Escherichia coli* ATCC 25922 and *Staphylococcus aureus* ATCC 29213 were used as control strains, and 73 strains of *B. cereus* were divided into sensitive (S), intermediary (I), and resistant (R).

### 2.5. Antibiotic Resistance Gene Analysis

The Comprehensive Antibiotic Research Database (CARD) contains all resistance information from the Antibiotic Genetic Database, with a data sharing platform. Users can upload antibiotic-related information at any time, achieving the real-time updating of data and ensuring the validity of the data, and this database is commonly used for researching resistance genes to date [46]. The whole-genome information of 73 strains of *B. cereus* was compared with that of the reference sequences in the CARD database using BLAST+ (https://ftp.ncbi.nlm.nih.gov/blast, accessed on 20 February 2023) to obtain information related to antibiotic-resistant genes of the strains, and the results were classified and counted.

## 3. Results

### 3.1. Chromogenic Medium Identification of B. cereus

The suspected *B. cereus* was collected from the Chromagar *B. cereus* chromogenic medium in accordance with the colony color and morphology, and the typical colony morphology. The results showed that 59 suspected *B. cereus*-positive samples were detected from 273 food samples, and 84 suspected *B. cereus* strains in total were isolated.

### 3.2. The 16S rRNA Sequencing and Whole Gene Sequencing Identification of B. cereus

The 16S rRNA sequences were uploaded to NCBI for BLAST comparison. The comparison results showed that among the 84 suspected *B. cereus* strains isolated from the samples, 73 strains with a sequence homology greater than 99% and published *B. cereus* strains in NCBI were determined to be positive. The remaining 11 suspected *B. cereus* strains had greater than 99% homology with other species in the database (six *Aspergillus*, three *Lactococcus lactis*, and two *Fusobacterium* spp.), and such strains were determined to be *B. cereus*-negative. Therefore, of the 273 samples collected, 56 (20.51%) were positive for *B. cereus*, and 73 strains of *B. cereus* in total were isolated. The results of whole-genome sequencing analysis showed that the 73 isolates were *B. cereus*.

### 3.3. Resistance Phenotype of B. cereus

The statistics of antibiotic resistance rate of 73 strains of *B. cereus* are shown in Figure 1. The resistance of 73 strains of *B. cereus* to different antibiotics was markedly different. The highest rate of resistance was found for *β*-lactam and sulfonamide antibiotics, both at 95.89%. Among the five antibiotics in the *β*-lactam group, all strains had a resistance rate of more than 70%, and cefotaxime was the most serious, with a resistance rate of 94.52%. The cefoxitin resistance rate was relatively low at 78.08%, but intermediate strains had the highest rate of 15.07%. Ticarcillin-Clavulanic acid had the lowest resistance of 73.97% among the *β*-lactams, but the number of strains susceptible to this antibiotic was the highest, with a ratio of 21.92%. Sulfonamides include two antibiotics with more pronounced differences in resistance, trimethoprim/sulfamethoxazole, with a resistance rate of 95.89% but a rate of sensitive strains of 4.11%, and sulfisoxazole, with a resistance rate of 28.77% and a rate of sensitive strains of 69.86%.

The 73 strains of *B. cereus* had low resistance to other antibiotics. The resistance rate of quinolone antibiotics was 16.44%, among which the resistance rate of ciprofloxacin was relatively high at 15.07%, the ratio of sensitive strains was 84.93%, and no intermediary strains were found. The rate of nalidixic acid resistance was 2.74%. In addition, 6.89% of the strains showed intermediaries, and the rate of sensitive strains was 90.41%. Carbapenem and aminoglycoside resistance was relatively low, at 2.74% and 4.11%, respectively. Among the aminoglycosides, no strains resistant to gentamicin were found, and the sensitivity rate was 100%. Streptomycin and kanamycin resistance rates were 2.74% and 1.37%, respectively, and both antibiotics had partial intermediary strains (5.48% and 8.22%). All strains were non-resistant to chloramphenicol, and the number of strains showing sensitivity reached 97.26%.

### 3.4. Comparison of Antibiotic Resistance of B. cereus in Different Regions

The antibiotic resistance rate of *B. cereus* isolated from seven regions was counted, and the results are shown in Table 1. Differences in the antibiotic resistance of *B. cereus* isolates were observed from different regions. Among them, *B. cereus* isolated from region A, region B, and region D had the highest resistance rates to ceftazidime, ampicillin, cefotaxime, and trimethoprim/sulfamethoxazole. Region C isolates showed the highest resistance rates to cefotaxime and trimethoprim/sulfamethoxazole. The strains isolated from region E had the highest resistance rate to ceftazidime and cefotaxime. The isolates from region F had the highest resistance rates to cefothiophene, ampicillin, amoxicillin, and trimethoprim/sulfamethoxazole. Regional G isolates had the highest resistance rate to trimethoprim/sulfamethoxazole. All regional isolates of *B. cereus* were sensitive to chloramphenicol and gentamicin.

### 3.5. Comparison of Antibiotic Resistance in Different Foodstuffs of B. cereus

The analysis of *B. cereus* isolated from six types of foods revealed that the antibiotic resistance of *B. cereus* in different foods also differed. The results shown in Table 2 indicate that *B. cereus* isolated from wild mushrooms and fresh vegetables had the highest resistance rate to two antibiotics, namely cefotaxime and trimethoprim/sulfamethoxazole. The strains isolated from soybean products had the highest resistance rate to trimethoprim/sulfamethoxazole. In addition, the strains isolated from pickled vegetables had the highest resistance rates to ceftazidime, cefotaxime, and trimethoprim/sulfamethoxazole. The strains isolated from frozen foods had the highest resistance rates to ceftazidime, ampicillin, amoxicillin, and cefotaxime. The strains isolated from cereals were the most sensitive to antibiotics, and the highest rates of resistance to ticarcillin-clavulanic acid and cephalothin were found. Moreover, the rates of resistance to these two antibiotics did not reach 100% in the other five food groups. All six types of food were considered sensitive to chloramphenicol and gentamicin.

### 3.6. Multi-Antibiotic Resistance Profile Analysis

The strains resistant to three or more antibiotics were considered as multi-antibiotic-resistant strains. The multi-antibiotic resistance spectrum of 73 strains of *B. cereus* resistant to 15 antibiotics is shown in Table 3. Among the 73 strains of the isolated *B. cereus*, only 2 strains were non-resistant to 15 antibiotics, whereas the rest of the strains were multi-resistant, and the ratio of 3 or more resistant strains reached 94.52%. The multi-antibiotic resistant types of the 15 antibiotics were mainly concentrated in 7-resistant and 8-resistant strains, and the total number of resistant strains was 36 and 15, respectively. The resistance spectrum of six antibiotic resistance was the largest, with seven types, and the resistance ratio of 73 isolates to six resistance reached 89.04%. The rate of eight or more resistant strains was 24.66%, and two 9-resistant and one 10-resistant strains were also included.

The resistance profiles of 73 *B. cereus* strains to six major classes of antibiotics are shown in Table 4. Among them, *β*-lactams are the main type of resistance, and are present in almost all resistance profiles. The three-fold resistance rate was 17.81%; among them, 10 isolates were resistant to *β*-lactams-Quinolones-Sulfonamides.

### 3.7. Antibiotic Resistance Genotype Analysis

A BLAST comparison of resistance genes of 73 strains of *B. cereus* was performed using the CARD database, and the distribution of resistance genes is shown in Table 5. Of the six major classes of antibiotics identified via resistance phenotyping, three classes of antibiotic resistance genes were identified. Among them, *β*-lactams have four antibiotic resistance gene profiles, of which *bla* and *bla2* are the main genotypes, and all strains carry these two genes. One genotype of chloramphenicol class was *catA* (50.68%), and no resistance to chloramphenicol was found in the antibiotic sensitivity test. Four genotypes of aminoglycosides are identified, of which *ant* (6) is the pre-dominant genotype (21.92%). Carbapenems, quinolones, and sulfonamides with antibiotic resistance detected via the antibiotic sensitivity test were not detected as antibiotic-resistant genes.

In addition, the presence of other resistance genes was identified from the genome-wide information of 73 *B. cereus* strains, including macrolides with four genotypes, primarily *abc-f* (98.63%). Only one genotype (*fosB*) was identified in fosfomycins, but all strains carried this genotype. Two genotypes of lincosamides were found, mainly *lsa* (15.07%). Streptomycins had three genotypes, in which the vat gene was also detected in all strains; glycopeptides had four genotypes, mainly *vanR-A* (13.7%) and *vanS-Pt* (13.7%); and tetracyclines also contained four genotypes, in which *tet* (10.96%) was the main genotype.

### 3.8. Correlation Analysis of Antibiotic-Resistant Phenotype and Genotype

The coincidence of an antibiotic-resistant genotype and antibiotic-resistant phenotype predicted via the whole-genome sequencing of *B. cereus* is shown in Table 6. Remarkable differences in correspondence were observed among the different types of antibiotic-resistant genes and phenotypes. The 73 strains of *B. cereus* had an overall conformity rate of 50.53% between antibiotic-resistant phenotypes and genes for six major classes of antibiotics. The total conformation rate between the *β*-lactam antibiotic-resistant phenotype and genotype was 95.89%, which was the highest among that of the six major classes of antibiotics. The compliance rate of strains that did not carry resistance genes with antibiotic-resistant phenotypes was 81.97%, and the overall compliance rate was 63.1%. The lowest overall compliance rate for sulfonamide antibiotics was 2.1%. The coincidence rate of chloramphenicols-, carbapenem-, quinolone-, and sulfonamide-resistant phenotypes with antibiotic-resistant genes was 0.

## 4. Discussion

The results of this study indicated remarkable differences in the antibiotic resistance of *B. cereus* isolated from various regions of a province. The highest level of resistance was observed for *β*-lactams and sulfonamide antibiotics, with resistance rates above 95%. *β*-lactam antibiotics are broad-spectrum antibiotics that are widely used in clinical treatment, resulting in serious antibiotic resistance [47]. Recent findings have shown that resistance to *β*-lactam antibiotics in *B. cereus* has become a common situation [29]. *B. cereus* isolated from retail aquatic products was extremely resistant to *β*-lactam antibiotics [48], which is consistent with the results of this study. In total, 39 strains of *B. cereus* isolated from ready-to-eat foods in Korea were sensitive to most of the antibiotics tested but highly resistant to *β*-lactam antibiotics [49].

Sulfonamides are also considered old antibiotics, and they are widely used in animal husbandry and aquaculture. Researchers have found a large number of sulfonamide-resistant microorganisms in the farm environment, livestock, and excreta [50,51]. Antibiotic studies of *B. cereus* isolated from raw foods showed that 24 strains of *B. cereus* were 100% resistant to sulfonamides [52]. The study of the antibiotic resistance of fish-derived *B. cereus* isolated from yellow-head catfish meat showed resistance to penicillin, ceftriaxone, oxacillin, ampicillin, ceftazidime, carbenicillin, cefuroxime, cefazolin, and Trimethoprim/Sulfamethoxazole [53], which was consistent with the results of the present study.

In a study of antibiotic resistance in *B. cereus* isolated from different regions, a geographical variation was found in *B. cereus*, leading to geographical variations in antibiotic resistance probably because of the frequency of antibiotic use within the region. The antibiotic resistance phenotyping of *B. cereus* isolated from different regions of China revealed that 238 strains of *B. cereus* were 100%, 100%, and 87.1% sensitive to gentamicin, vancomycin, and clindamycin, respectively [54]. The antibiotic resistance of *B. cereus* isolated from different food groups also varied with the resistance rate of *B. cereus* isolated from fresh vegetables, which was generally higher than that of *B. cereus* isolated from other foods. This result may be due to the presence of *B. cereus* antibiotic contamination in the soil environment when the fresh vegetables were grown, which was then carried over to the vegetables because of poor hygiene. The contamination status of antibiotics in vegetable soils in Yangling District, Shaanxi Province, was assessed; the detection rate of quinolone, sulfonamide, and tetracycline antibiotics in the soils of 20 planting bases was 100%, and the detection rate of macrolide antibiotics was 62% [19].

Multi-antibiotic-resistant strains refer to bacteria that are resistant to three or more antibiotics. At present, methicillin-resistant *Staphylococcus aureus*, carbapenem-resistant *Enterobacteriaceae*, and carbapenem-resistant *Pseudomonas aeruginosa* are being monitored in clinical practice. The multi-antibiotic resistance of *B. cereus* has become a serious public health problem that jeopardizes human health, and the investigation of its multi-antibiotic resistance is important. In the present study, while studying the isolated isolates of *B. cereus*, only two strains were non-resistant to 15 antibiotics, whereas the rest of the strains were resistant to 3 and more antibiotics, which amounted to a resistance rate of 94.52%. Therefore, the antibiotic resistance of *B. cereus* is serious, and this problem should be a matter of great concern to prevent the emergence of a large number of ultra-broad-spectrum-resistant strains and infections.

The 73 strains of *B. cereus*, as determined via partial antibiotic resistance gene detection, had a low compliance with the resistance phenotypic profile. Among them, sulfonamides were the most evident, with a coincidence rate of only 2.1%. Briefly, 95.89% of these isolates showed resistance to sulfonamide antibiotics in antibiotic sensitivity tests; however, no sulfonamide antibiotic-resistant genes were found in the actual tests. The mechanism of resistance of sulfonamide antibiotics is primarily through their interaction with an important folate synthase enzyme in bacterial growth, thereby inhibiting bacterial growth and reproduction, and achieving resistance. In contrast, resistance to sulfonamide antibiotics is primarily related to the mutated gene *sul* of folate synthase, which regulates the binding of folate synthase and sulfonamide antibiotics, thereby causing strains to produce resistance. The expression of *sul* genes in microorganisms was found to be time-specific, with both strains tested, with or without a sulfonamide antibiotic-resistant phenotype, showing the highest expression of *sul* genes at one point in time, followed by a rapid decline to a very low level [51]. However, the exact mechanism of action that produces these genes is currently unknown, which may explain the undetected sulfonamide-resistant gene. In addition, chloramphenicol-resistant phenotypes were found for resistance cases, but the results of resistance gene testing revealed that 37 strains carried chloramphenicol-resistant genes, which may be due to resistance gene silencing. An analysis of the resistance of *lactic acid bacteria* to antibiotics revealed that some strains were resistant to ampicillin, penicillin, chloramphenicol, and tetracycline, but no resistance genes were detected. Some other strains had a cat gene mediating chloramphenicol resistance, but none of these strains had phenotypic resistance to chloramphenicol, confirming the occurrence of resistance gene silencing [55]. In addition to these six major classes of antibiotics, genome-wide analysis also detected the presence of many other classes of resistance genes, demonstrating that the isolated *B. cereus* may also be at risk of resistance to other antibiotics.

## 5. Conclusions

In this study, 73 strains of *B. cereus* were isolated and purified from six kinds of commercially available plant foods in seven regions of a province. Resistant phenotypes of 73 *B. cereus* isolates were identified using the K-B paper diffusion method provided by CLSI. The results of antibiotic sensitivity tests showed that the overall antibiotic resistance of 73 strains of *B. cereus* was serious, except for that against gentamicin and chloramphenicol. All strains had different degrees of resistance to 13 other antibiotics, among which resistance to *β*-lactam and sulfonamide antibiotics was the most serious, and a large number of multi-antibiotic-resistant strains were found. Bioinformatic methods were used to analyze the expression of antibiotic-resistant genes in *B. cereus*. The results of antibiotic-resistant gene detection showed that three types of antibiotic-resistant genes were found to be associated with six types of antibiotics identified via an antibiotic-resistant phenotype. In addition, the presence of other resistant genes was found from the whole-genome information, indicating that *B. cereus* has a multi-antibiotic resistance potential for these antibiotics. The analysis of the correlation between antibiotic-resistant phenotypes and genotypes revealed remarkable differences among different types of antibiotic-resistant phenotype and genotype correspondence. This situation may be related to the resistance mechanism of different antibiotics. In this study, antibiotic resistance is not only present in *B. cereus* isolated from plant foods and edible wild mushrooms, but also in other bacteria that are present in other foods in daily life, which is a huge threat to human health. This study provides useful information on the presence of bacterial antibiotic resistance in various types of foods and highlights the importance of future research in this area.

## Figures and Tables

**Figure 1 microorganisms-11-02948-f001:**
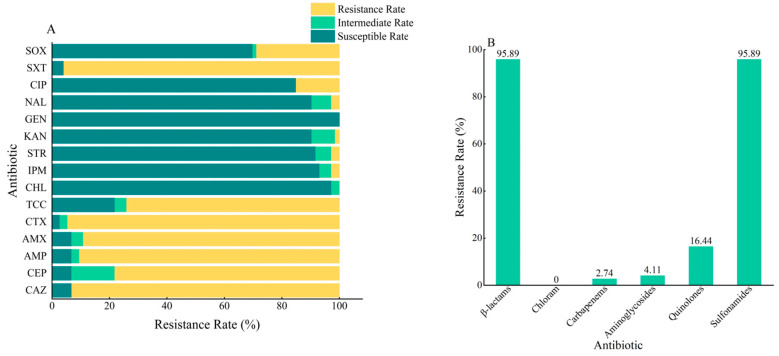
Antimicrobial resistance statistics for 73 *B. cereus*. (**A**) Resistance rate of 73 strains of *B. cereus* to 15 antibiotics; (**B**) resistance rate of 73 strains of *B. cereus* to 6 major classes of antibiotics.

**Table 1 microorganisms-11-02948-t001:** Antimicrobial resistance of *B. cereus* in different regions.

Antibiotic *	Resistance Rate (%)						
	Region A	Region B	Region C	Region D	Region E	Region F	Region G
Ceftazidime (CAZ)	90	100	80	95.45	100	80	91.67
Cephalothin (CEP)	90	78.57	80	72.73	80	100	66.67
Ampicillin (AMP)	90	100	80	95.45	80	100	75
Amoxicillin (AMX)	90	85.71	80	95.45	80	100	83.33
Cefotaxime (CTX)	90	100	100	95.45	100	80	91.67
Ticarcillin-Clavulanic acid (TCC)	60	64.29	40	86.36	80	80	83.33
Chloramphenicol (CHL)	0	0	0	0	0	0	0
Imipenem (IPM)	0	0	0	9.09	0	0	0
Streptomycin (STR)	0	7.14	0	0	0	20	0
Kanamycin (KAN)	0	0	0	0	0	0	8.33
Gentamicin (GEN)	0	0	0	0	0	0	0
Nalidixic acid (NAL)	0	0	0	4.55	0	0	8.33
Ciprofloxacin (CIP)	20	28.57	40	9.09	20	0	0
Trimethoprim/Sulfamethoxazole (SXT)	90	100	100	95.45	80	100	100
Sulfisoxazole (SOX)	20	28.57	20	18.18	20	60	50

* Complete name of antibiotics (antibiotic abbreviations).

**Table 2 microorganisms-11-02948-t002:** Antimicrobial resistance of *B. cereus* in different foods.

Antibiotic	Resistance Rate (%)					
	Wild Mushroom	Soybean Products	Fresh Vegetables	Preserved Vegetable	Frozen Food	Cereals
CAZ	90.91	90	93.33	94.74	92.86	100
CEP	72.73	50	93.33	73.68	85.71	100
AMP	90.91	80	100	89.47	92.86	75
AMX	81.82	80	93.33	89.47	92.86	100
CTX	100	90	100	94.74	92.86	100
TCC	63.64	60	80	73.68	78.57	100
CHL	0	0	0	0	0	0
IPM	0	0	6.67	5.26	0	0
STR	0	0	6.67	0	7.14	0
KAN	0	10	0	0	0	0
GEN	0	0	0	0	0	0
NAL	0	0	0	10.53	0	0
CIP	18.18	10	13.33	15.79	21.43	0
SXT	100	100	100	94.74	85.71	100
SOX	18.18	40	26.67	47.37	7.14	25

**Table 3 microorganisms-11-02948-t003:** Multi-resistance profile of 73 *B. cereus* to 15 antibiotics.

Type of Resistance	Resistant Spectrum	Number of Isolates	Proportion (%)
0	—	2	2.74%
2	K-SXT	1	2.74%
	CTX-SXT	1	
3	CAZ-CTX-SXT	1	2.74%
	CF-AMP-AMX	1	
4	CAZ-AMP-CTX-SXT	1	1.37%
5	CAZ-AMP-AMX-CTX-SXT	1	1.37%
6	CAZ-AMP-AMX-CTX-TIC-SXT	5	15.07%
	CAZ-CF-AMP-AMX-CTX-SXT	1	
	CAZ-AMP-CTX-TIC-SXT-SF	1	
	CAZ-AMX-CTX-TIC-SXT-SF	1	
	CAZ-CF-AMP-CTX-SXT-SF	1	
	CAZ-CF-AMP-AMX-CTX-CIP	1	
	CAZ-CF-AMX-CTX-TIC-SXT	1	
7	CAZ-CF-AMP-AMX-CTX-TIC-SXT	26	49.31%
	CAZ-CF-AMP-AMX-CTX-CIP-SXT	8	
	CAZ-AMP-AMX-CTX-TIC-SXT-SF	1	
	CAZ-CF-AMP-AMX-CTX-TIC-CIP	1	
8	CAZ-CF-AMP-AMX-CTX-TIC-SXT-SF	13	20.55%
	CAZ-AMP-AMX-CTX-TIC-NA-SXT-SF	1	
	CAZ-CF-AMP-AMX-CTX-TIC-S-SXT	1	
9	CAZ-CF-AMP-AMX-CTX-TIC-IPM-SXT-SF	2	2.74%
10	CAZ-CF-AMP-AMX-CTX-TIC-NA-CIP-SXT-SF	1	1.37%

**Table 4 microorganisms-11-02948-t004:** Multi-resistance profile of 73 *B. cereus* to 6 classes of antibiotics.

Type of Resistance	Resistant Spectrum	Number of Isolates	Proportion (%)
0	-	2	2.74%
2	*β*-lactams—Sulfonamides	54	79.45%
	*β*-lactams—Aminoglycosides	1	
	*β*-lactams—Quinolones	2	
	Aminoglycosides—Sulfonamides	1	
3	*β*-lactams—Quinolones—Sulfonamides	10	17.81%
	*β*-lactams—Carbapenems—Sulfonamides	2	
	*β*-lactams—Aminoglycosides—Sulfonamides	1	

**Table 5 microorganisms-11-02948-t005:** Results of antimicrobial resistance gene detection in 73 *B. cereus*.

Antibiotic Category	Resistance Gene Spectrum	Number of Isolates	Detection Rate (%)
*β*-lactams	*bla*, *bla2*	73	100
	*bla*, *bla2*, *blaTEM*	1	1.37
	*bla*, *bla2*, *hugA*	3	4.11
	*bla*, *bla2*, *blaOXA*	1	1.37
Chloramphenicols	*catA*	37	50.68
Aminoglycosides	*aph*(3′)*-Iia*	4	5.48
	*Ant* (6)	16	21.92
	*aadA1*, *aadA31*	2	2.74
	*ant* (6), *ant* (4′)*-I*	1	1.37
Macrolides	*abc-f*	72	98.63
	*mphL*	8	10.96
	*abc-f*, *mphL*	7	9.59
	*abc-f*, *msr*, *cfr*	1	1.37
Fosfomycins	*fosB*	73	100
Lincosamides	*lsa*	11	15.07
	*cfr*	1	1.37
Streptomycins	*vat*	73	100
	*vat*, *lsa*	12	16.44
	*vat*, *cfr*	1	1.37
Glycopeptides	*vanR-A*	10	13.7
	*vanR*, *vanS*	3	4.11
	*vanR-A*, *vanS-Pt*, *vanY*	3	4.11
	*vanR-A*, *vanS-Pt*	7	9.59
Tetracyclines	*tet*	8	10.96
	*tet(H)*	1	1.37
	*tet_rib_protect*	2	2.74
	*tet(H)*, *tet_rib_protect*	1	1.37

**Table 6 microorganisms-11-02948-t006:** Conformity of antimicrobial resistance phenotypes and genotypes.

Antibiotic Category	Sensitive Strain			Drug Resistant Strain			Total Rates (%)
	Sensitivity	Not Carrying Drug-Resistant Genes	Coincidence Rates (%)	Drug Resistance	Carrying Resistance Genes	Coincidence Rates (%)	
*β*-lactams	0	0	100	70	73	95.89	95.89
Chloramphenicols	71	36	50.7	0	37	0	33.33
Carbapenems	68	73	93.15	2	0	0	90.67
Aminoglycosides	61	50	81.97	3	23	13.04	63.1
Quinolones	57	73	78.08	12	0	0	67.06
Sulfonamides	3	73	4.11	70	0	0	2.1
Total	-	-	-	-	-	-	50.53

## Data Availability

Data are contained within the article.

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
