# Peer review of "Antibiotic Resistance of Bacillus cereus in Plant Foods and Edible Wild Mushrooms in a Province"

_microorganisms, 2023, doi:10.3390/microorganisms11122948_

Round 1

Reviewer 1 Report

Comments and Suggestions for Authors

The research paper investigated the antibiotic resistance profile of Bacillus cereus strains isolated from several regions of a Chinese province. The genomic study showed the presence of resistance genes, providing the strains with multiple-resistance. The results of this study help bring more information to the complex mechanism of bacterial resistance.

Comments:

- The authors chose not to give the specific name of the province and regions studied. I highly recommend specifying the study location cause it will help other researchers who wish to investigate the relationship between resistance and location (which was found in this study but not thoroughly discussed).

- Figure 2A and Table 2: The data are repeated. Please choose one way to put the data in your manuscript.

- The tables and figures need to be self-explanatory. Please add the antibiotic abbreviations in the footnote. I recommend using the complete name of the antibiotics when it's possible (like tables 2, 3, 4) to avoid a long footnote.

- Please add references to tables (3 and more) in the text replacing "Error! Reference source not found"

- Lines 190-192: Please delete the figure's title.

- Lines 213-214: "Ceftazidime and cefotaxime isolated from region E had the highest resistance rate". This sentence is incorrect. Please revise the results section. 

- Lines 222-223: "(isolated from wild mushrooms and fresh vegetables)": This is an essential part of the sentence, please delete the parentheses. 

- Table 5: the column "proportion" is confusing. When reading the line "type of resistance 2" for example, you expect to see the proportion of strains possessing resistance to 2 antibiotics (here it's 2 strains), but the proportion reflects 2 or more (97.26%). Please make it clearer for readers. Same remark for table 6.

Comments on the Quality of English Language

Generally, the manuscript is well-written. The authors are encouraged to do a last revision to fix some small mistakes.

Please verify some split words in the middle of sentences, like an-tibiotic (line 189).

Please format the species' names in italics.

Author Response

Response to Reviewer 1' Comments

1: The authors chose not to give the specific name of the province and regions studied. I highly recommend specifying the study location cause it will help other researchers who wish to investigate the relationship between resistance and location (which was found in this study but not thoroughly discussed).

Answer: Thank you for your valuable suggestion. We understand your suggestion, your opinion is that specifying the study location will help other researchers who wish to investigate the relationship between resistance and location. Actually, we have some concerns about disclosing the sampling location. Due to the particularity of the funded project and research content, it is not convenient to disclose the sampling location, and then we choose to use the terms of a province and a region. But all the bacteria we obtained from this research are already stored in our culture bank, and if researchers need to use certain strains, we can provide them.

2: Figure 2A and Table 2: The data are repeated. Please choose one way to put the data in your manuscript.

Answer: Thank you for your suggestion. We used the figure 2A in the manuscript, and the Table 2 have been deleted.

3: The tables and figures need to be self-explanatory. Please add the antibiotic abbreviations in the footnote. I recommend using the complete name of the antibiotics when it's possible (like tables 2, 3, 4) to avoid a long footnote.

Answer: Thank you for your suggestion. We have revised the tables as your suggestion, the abbreviation of antibiotics in Table1 is transformed into the complete name of antibiotics. For the simplicity of the tables, especially the multi-resistance profile of 73 B. cereus to 15 antibiotics in Table 3, considering the complete names of the antibiotics are too long, so we used the abbreviations of the antibiotics.

4: Please add references to tables (3 and more) in the text replacing "Error! Reference source not found".

Answer: Thanks for your suggestion. In the result part, these errors were due to a problem with the quote of the tables and pictures, and we have revised and typeset the tables and pictures. (Line 182-183, Line 210, Line 224, Line 240, Line 250, Line 256, and Line 275)

5: Lines 190-192: Please delete the figure's title.

Answer: Thank you for your suggestion. We delete figure's title. (Line 182-183)

6: Lines 213-214: "Ceftazidime and cefotaxime isolated from region E had the highest resistance rate". This sentence is incorrect. Please revise the results section.

Answer: Thank you for your comment. We have revised as your suggestion. "Ceftazidime and cefotaxime isolated from region E had the highest resistance rate "revised as “The strains isolated from region E had the highest resistance rate to ceftazidime and cefotaxime”. (Line 214-215)

7: Lines 222-223: "(isolated from wild mushrooms and fresh vegetables)": This is an essential part of the sentence, please delete the parentheses.

Answer: Thank you for your suggestion. We have revised as your suggestion. (Line 224-226)

8:Table 5: the column "proportion" is confusing. When reading the line "type of resistance 2" for example, you expect to see the proportion of strains possessing resistance to 2 antibiotics (here it's 2 strains), but the proportion reflects 2 or more (97.26%). Please make it clearer for readers. Same remark for table 6.

Answer: Thank you for your suggestion. We have revised as your suggestion, the proportion column was recalculated, and the results showed that the number of strains with corresponding antibiotic types accounted for the proportion of drug resistance of all isolates. (Table 3 and 4)

  1. Please format the species' names in italics.

Answer: Thank you for your suggestion. We have changed the species name to italic. (Line152, Line171-180, Line183, Line206, Lin 222-225, Line 267, Line272, Line293, Line 321-322, Line339-343, Line 348-349)

Reviewer 2 Report

Comments and Suggestions for Authors

The manuscript submitted for review describes the antibiotic resistance profiles of Bacillus cereus isolated from food. These bacteria are not the primary focus of clinical microbiologists, and their resistance profiles still hold many mysteries. Strains from food are important as they directly interact with consumers. However, the paper needs significant improvement.

Introduction: This section is excessively long. Describing potential toxicity is unjustified since the authors do not investigate toxins, determine toxicity profiles, or identify pathogenic bacterial types. Moreover, they fail to reference recent antibiotic studies indicating, for example, different resistance profiles of B. cereus from animal-origin and plant-origin foods.

Materials and Methods: The description is sufficient and clear. However, there is a question about the classification of fungi as plants, as implied in the paper (lines 96-97).

General Comment: Why do the authors identify B. cereus without considering other group representatives? B. cereus, B. thuringiensis, and B. mycoides have over 99% homology in 16S rRNA. Why not analyze other features, such as cry genes or colony morphology using phase-contrast microscopy? In my opinion, each strain should be re-evaluated according to the methodology accepted for bacteria in this group or assigned to the collective group Bacillus cereus sensu lato.

Results: The presented results are interesting and supported by other studies. Their credibility is not in question. With 16S rRNA sequencing data, a dendrogram showing phylogenetic relationships between strains could enhance the discussion. The image of Petri dishes adds nothing to the manuscript and can be removed without harming it.

I believe presenting regions using a map would be more illustrative (paragraph 3.4.). In line 235, the authors provide a definition of multidrug resistance that is inconsistent with the classical definition referring to at least three classes of antibiotics. Please provide an explanation or correction. The large number of tables overwhelms and complicates manuscript reading.

Discussion: How does the presence of antibiotics in the soil affect resistance? Was the level of soil contamination with antibiotics determined, or was their presence only confirmed? What is the nature of antibiotic use in the soil (treatment of humans, animals, growth promoters)?

Overall, I believe the study has potential, and the results may be interesting, but certain elements highlighted in the review need improvement. I also lack information indicating the potential significance of horizontal gene transfer in the development of antibiotic resistance. The geographic correlation is not illustrative, clear phylogenetic data and precise strain identification are missing. Additionally, the exclusion of bacteria psychrotolerant at 36 degrees Celsius is noteworthy, considering their significant presence in the soil.

Author Response

Response to Reviewer 2' Comments

1: Introduction: This section is excessively long. Describing potential toxicity is unjustified since the authors do not investigate toxins, determine toxicity profiles, or identify pathogenic bacterial types. Moreover, they fail to reference recent antibiotic studies indicating, for example, different resistance profiles of B. cereus from animal-origin and plant-origin foods.

Answer: Thanks for your suggestion. Your suggestion is valuable and informative! We have deleted the part of potential toxicity and add the content about different resistance of Bacillus cereus in plant-origin foods in the section of introduction (lines 61-77).

2: Materials and Methods: The description is sufficient and clear. However, there is a question about the classification of fungi as plants, as implied in the paper (lines 96-97).

Answer: Thanks for your suggestion. When we named the title we neglected to mention that wild mushrooms belong to the fungi group. We have changed the title into Antibiotic Resistance of Bacillus cereus in Plant Foods and Edible Wild Mushrooms in a Province. (lines 1-3, lines 84-86, lines 89-90)

3: General Comment: Why do the authors identify B. cereus without considering other group representatives? B. cereus, B. thuringiensis, and B. mycoides have over 99% homology in 16S rRNA. Why not analyze other features, such as cry genes or colony morphology using phase-contrast microscopy? In my opinion, each strain should be re-evaluated according to the methodology accepted for bacteria in this group or assigned to the collective group Bacillus cereus sensu lato.

Answer: Thanks for your suggestion. Your suggestion is valuable. The three strains of Bacillu. cereus, B. thuringiensis, and B. mycoide share many of the same phenotypic characteristics, and the 16S rRNA sequence data showed that the homology was 99%[1]. The B. cereus group has always been a controversial topic [2]. Firstly, B. cereus was isolated and identified by B. cereus chromogenic medium, and then 16S rRNA homology analysis was performed. In order to ensure the rigor of the experiment, we consulted relevant literature on the differentiation of B. cereus and found that many studies used whole genome sequencing to identify the molecular biology of B. cereus [3], and genetic identification by sequencing technology could reduce the false positive results [2]. This method provides the best means for the typing and genetic identification of B. cereus [4]. Based on this, we conducted whole genome sequencing to further confirm that the isolated strain was B. cereus. As you proposed the existence of specific cry gene is also a good method to identify three members. Meanwhile, it was also mentioned in the literature, it was also a good method to identify three members of the B. cereus group: B. cereus sensu stricto, B. mosaicus and B. thuringiensis through the analysis of the whole genome sequencing data [5]. Based on the literature we consulted before, we conducted whole genome sequencing and finally obtained the result that the isolated strain was B. cereus. Due to the negligence of our writing, manuscript did not explain clearly the final results obtained by whole genome sequencing. The whole genome sequencing part was supplemented in the material and the relevant literature was cited. (Line 106, Line 126-127, Line 171, Line179-180)

[1] Fox, G.E.; Wisotzkey, J.D.; Jurtshuk, P. How Close Is Close: 16S rRNA Sequence Identity May Not Be Sufficient To Guarantee Species Identity. International Journal of Systematic Bacteriology. 1992, 42, 166–170.

[2] Léonard, C.; Chen, Y.; Mahillon, J. Diversity and Differential Distribution of IS231, IS232 and IS240 among Bacillus Cereus, Bacillus Thuringiensis and Bacillus Mycoides. Microbiology. 1997, 143, 2537–2547.

[3] Zhang, H.; Liu, X.W.; Gu, Q.F.; Chang, Z.Y.; Zhu Y.Q.; Zhang X. Molecular characteristics and antibiotic resistance of Bacillus cereus from foods using whole genome sequencing. Chinese Journal of food hygiene. 2021, 529–535.

[4] Fei, P.; Yuan, X.; Zhao, S.; Yang, T.; Xiang, J.; Chen, X.; Zhou, L.; Ji, M. Prevalence and Genetic Diversity of Bacillus Cereus Isolated from Raw Milk and Cattle Farm Environments. Curr Microbiol. 2019, 76, 1355–1360.

[5] Fraccalvieri, R.; Bianco, A.; Difato, L.M.; Capozzi, L.; Del Sambro, L.; Simone, D.; Catanzariti, R.; Caruso, M.; Galante, D.; Normanno, G.; et al. Toxigenic Genes, Pathogenic Potential and Antimicrobial Resistance of Bacillus Cereus Group Isolated from Ice Cream and Characterized by Whole Genome Sequencing. Foods. 2022, 11, 2480-2492.

4: Results: The presented results are interesting and supported by other studies. Their credibility is not in question. With 16S rRNA sequencing data, a dendrogram showing phylogenetic relationships between strains could enhance the discussion. The image of Petri dishes adds nothing to the manuscript and can be removed without harming it.

Answer: Thanks for your suggestion. The 16S rRNA data used to generate a dendrogram showing phylogenetic relationships involves genetic and bacterial traceability studies, which we analyze and study in depth in another article. According to your suggestion, the removed image of Petri dishes has no effect to the manuscript, and we have deleted it.

5: I believe presenting regions using a map would be more illustrative (paragraph 3.4.). In line 235, the authors provide a definition of multidrug resistance that is inconsistent with the classical definition referring to at least three classes of antibiotics. Please provide an explanation or correction. The large number of tables overwhelms and complicates manuscript reading.

Answer: Thank you for your valuable suggestions. We understand your suggestion, presenting regions using a map would be more illustrative. However, we have some concerns about disclosing the sampling location. Due to the particularity of the funded project and research content, it is not convenient to disclose the sampling location, and then we choose to use the terms of a province and a region. But all the bacteria we obtained from this research are already stored in our culture bank, and if researchers need to use certain strains, we can provide them.

We have corrected the definition of multi-drug resistance by searching for information. (http://www.nhc.gov.cn/cms-search/xxgk/getManuscriptXxgk.htm?id=50487) (Line 238-239, Line 240-242, Line 251-252).

Regarding the large number of tables, we have collated and deleted some tables.

6: Discussion: How does the presence of antibiotics in the soil affect resistance? Was the level of soil contamination with antibiotics determined, or was their presence only confirmed? What is the nature of antibiotic use in the soil (treatment of humans, animals, growth promoters)?

Answer: Thank you for your suggestions. In the discussion, we suggested that that the reason why isolates from different regions and different samples develop different antibiotic resistance may be related to the soil environment. The focus of our study was on the resistance of Bacillus cereus in sampled foods. By consulting the relevant literature, it is known that the reason for the antibiotic resistance of the strain may be given by the soil during the growth process, it may also be polluted by natural environment and human activities after picking, or it may be the secretion and metabolism of the bacteria itself. We only put forward the possible reasons for the results from the perspective of food safety, and provide reference for future related research.

Consult the relevant literature about your suggestion “how does the presence of antibiotics in the soil affect resistance”, we believe that antibiotic resistance is only affected when antibiotic pollution in soil reaches a certain level. The content of natural antibiotics in soil is very low, but it can produce certain selection pressure, including antibiotics, organic fertilizers, animal manures, digestates or sewage sludge, heavy metals, quaternary ammonium compounds and biocides, on microbial organisms. It endows microorganisms with resistance, so that antibiotic resistance naturally exists in soil microorganisms [1]. The abuse of antibiotics is a key factor leading to the increase of antibiotic resistance abundance. For example, the long-term and frequent application of tetracycline in livestock and poultry breeding and the prevention and treatment of human diseases has led to the increasing resistance of tetracycline in the environment [2].

Irrational use of antibiotics in livestock farms is an important cause of resistance gene development, therefore, the application of animal manure or compost in the soil will introduce exogenous antibiotic resistance genes (ARGs) into the soil [3]. Antibiotic residues in feces will be a positive selection pressure to enrich soil resistance [4].

Human activities can accelerate the distribution and spread of ARGs in the soil environment [5]. It can be summarized into two aspects: fertilizer application and wastewater or reclaimed water irrigation and reuse [6].

The transmission process of antibiotic resistance from soil to plant is mainly described as follows: (1) "Soil-root system" is the main pathway for ARGs to enter plants; (2) Root exudation is an important process determining plant interactions with the soil environment. The soil-plant system consists of the soil environment around plants and their root systems, and ARGs can be transferred to plants along with microbial transfer in the "soil-root" or "soil-air-leaf" system, and passed to humans through the food chain [7]; (3) Root system morphology and structure influence the spread of antibiotic-resistant bacteria from soil to plants[8]; (4) Horizontal transfer of genes in the inter-root environment promotes the spread of ARGs through the soil-root system[9]; (5)“Soil-air-phyllosphere” is the main way for ARGs in soil to transfer to the phyllosphere; (6) The distribution of ARGs in plants is universal and diverse, and can exist in tissues and organs such as roots, stems, leaves, fruits and seeds ARGs[10].

[1] Nguyen, B.-A.T.; Chen, Q.-L.; He, J.-Z.; Hu, H.-W. Microbial Regulation of Natural Antibiotic Resistance: Understanding the Protist-Bacteria Interactions for Evolution of Soil Resistome. Science of The Total Environment. 2020, 705, 135882-135892.

[2] Hellweger, F.; Ruan, X.; Sanchez, S. A Simple Model of Tetracycline Antibiotic Resistance in the Aquatic Environment (with Application to the Poudre River). IJERPH. 2011, 8, 480–497.

[3] McKinney, C.W.; Dungan, R.S.; Moore, A.; Leytem, A.B. Occurrence and Abundance of Antibiotic Resistance Genes in Agricultural Soil Receiving Dairy Manure. FEMS Microbiology Ecology. 2018, 94,

[4] Zhao, Y.; Su, J.-Q.; An, X.-L.; Huang, F.-Y.; Rensing, C.; Brandt, K.K.; Zhu, Y.-G. Feed Additives Shift Gut Microbiota and Enrich Antibiotic Resistance in Swine Gut. Science of The Total Environment. 2018, 621, 1224–1232.

[5] Han, X.-M.; Hu, H.-W.; Chen, Q.-L.; Yang, L.-Y.; Li, H.-L.; Zhu, Y.-G.; Li, X.-Z.; Ma, Y.-B. Antibiotic Resistance Genes and Associated Bacterial Communities in Agricultural Soils Amended with Different Sources of Animal Manures. Soil Biology and Biochemistry. 2018, 126, 91–102

[6] Zhang, N.; Li, M.; Liu, X. Distribution and transformation of antibiotic resistance genes in Soil. China Environmental Science. 2018, 2609–2617.

[7] Deng, B.; Li, W.; Lu, H.; Zhu, L. Film Mulching Reduces Antibiotic Resistance Genes in the Phyllosphere of Lettuce. Journal of Environmental Sciences. 2022, 112, 121–128.

[8] Herms, C.H.; Hennessy, R.C.; Bak, F.; Dresbøll, D.B.; Nicolaisen, M.H. Back to Our Roots: Exploring the Role of Root Morphology as a Mediator of Beneficial Plant–Microbe Interactions. Environmental Microbiology. 2022, 24, 3264–3272.

[9] Forsberg, K.J.; Patel, S.; Gibson, M.K.; Lauber, C.L.; Knight, R.; Fierer, N.; Dantas, G. Bacterial Phylogeny Structures Soil Resistomes across Habitats. Nature. 2014, 509, 612–616.

[10] Chen, Q.-L.; Cui, H.-L.; Su, J.-Q.; Penuelas, J.; Zhu, Y.-G. Antibiotic Resistomes in Plant Microbiomes. Trends in Plant Science. 2019, 24, 530–541.

Reviewer 3 Report

Comments and Suggestions for Authors

The concept of the study is good but have some major issues which need to be addressed before accepting the paper.

Line 85: In recent years foodborne diseases caused by B. cereus have been increasing annually, bringing serious impact on people's health and social development. Please provide with bibliographical date and reference.

Line 109-122: the sentences are repeated.

Line 136: Antibiotic resistance testing of isolate

Please redefine the whole used method with appropriate scientific rigor, clarify certain concepts, for example: antibiotic sensitized tablets, inhibition circle, the allergy test solid. Reference to the literature is required. 

Explain why you used data on Enterobacteroaceae? 

Figure 1. Typical colony morphology of some B. cereus strains

Species names should be written by italics consistently throughout the manuscript.

correct spelling errors, for example: Chloram Phenicols (in table 2), Lacto-coccus lactis (line 184). 

It appears in several places in the results section: Error! Reference source not found. Please correct! 

Author Response

Response to Reviewer 3' Comments

1: Line 85: In recent years foodborne diseases caused by B. cereus have been increasing annually, bringing serious impact on people's health and social development. Please provide with bibliographical date and reference.

Answer: Thank you for your suggestion. We provided the references to the phrase “In recent years foodborne diseases caused by B. cereus have been increasing annually, bringing serious impact on people's health and social development.” (Line 78-79)

2: Line 109-122: the sentences are repeated.

Answer: Thank you for your suggestion. Due to our oversight, some of the content was duplicated, and the duplicates have been removed. (Line 97-104)

3: Line 136: Antibiotic resistance testing of isolate.

Please redefine the whole used method with appropriate scientific rigor, clarify certain concepts, for example: antibiotic sensitized tablets, inhibition circle, the allergy test solid. Reference to the literature is required.

Answer: Thanks for your advice. The antibiotic sensitized tablets, inhibition circle, the allergy test solid have been redefined and references also added. (Line 129-132, Line 149-155)

5: Explain why you used data on Enterobacteroaceae?

Answer: Thanks for your advice. We used data on Enterobacteroaceae for the following reasons, firstly, Enterobacteriaceae and Bacillus cereus belong to bacillus, and then Enterobacter is easy to obtain pure strains with selective medium, and has no inherent resistance. Through literature review, the authors of the this reference used E.coli ATCC25922 as a control, so we were inspired to adopt the same method. (Line 152-153)

Fraccalvieri, R.; Bianco, A.; Difato, L.M.; Capozzi, L.; Del Sambro, L.; Simone, D.; Catanzariti, R.; Caruso, M.; Galante, D.; Normanno, G.; et al. Toxigenic Genes, Pathogenic Potential and Antimicrobial Resistance of Bacillus Cereus Group Isolated from Ice Cream and Characterized by Whole Genome Sequencing. Foods. 2022, 11, 2480-2492.

6: Species names should be written by italics consistently throughout the manuscript.

correct spelling errors, for example: Chloram Phenicols (in table 2), Lacto-coccus lactis (line 184).

It appears in several places in the results section: Error! Reference source not found. Please correct!

Answer: Thanks for your advice. We have corrected as your suggestion. (Table 5, 6, Line 176-177)

In the result part, these errors were due to a problem with the quote of the tables and pictures, and we revised and typeset the tables and pictures. (Line 181-182, Line 209, Line 223, Line 239, Line 249, Line 248-249, Line 255, Line 274)

Round 2

Reviewer 3 Report

Comments and Suggestions for Authors

Dear Authors!

Please find attached the comments.

Author Response

Response to Reviewer 3' Comments

1: Line 131: the Clinical and Laboratory Standards Institute” – repeats

Answer: Thanks for your suggestions. We have deleted the duplicate Clinical and Laboratory Standards Institute. (Line 131)

2: Line 138: “the antibiotic-sensitized tablets”- antibiotic disks is the correct term.

Answer: Thanks for your suggestions. We have modified “the antibiotic-sensitized tablets” to “antibiotic disks”. (Line 140)

3: Explain why you used data on Enterobacteroaceae?

Answer: Thanks for your suggestions. We have carefully read the references that you listed, and we have revised our manuscript, and used Staphylococcus aureus as the control strain for antibiotics sensitivity determination.

Previously, we ignored that Enterobacteriaceae is a Gram-negative bacterium. Therefore, according to your suggestion, we changed the reference strain into S. aureus.

At the same time, we consulted in the file of CLSI (Version 2013 and 2021), the antibiotic susceptibility test criteria of Enterobacteriaceae and S. aureus in chloramphenicol, aminoglycosides, quinolones and sulfonamides are consistent. However, there is no fixed criteria for β-lactams and carbapenems of S. aureus, but it is sensitive to these two types of antibiotics. We also consulted the reference (1), in this study, Escherichia coli ATCC ® 25922 and S. aureus ATCC ® 29213 were both adopted as the reference strains. Therefore, we classified the results of Enterobacteriaceae in the criteria of β-lactams and carbapenems as controls. (Line 148-154)

Thank you again for your suggestions, which is very meaningful to improve the rigor of our manuscript.

(1) Fraccalvieri, R.; Bianco, A.; Difato, L.M.; Capozzi, L.; Del Sambro, L.; Simone, D.; Catanzariti, R.; Caruso, M.; Galante, D.; Normanno, G.; et al. Toxigenic Genes, Pathogenic Potential and Antimicrobial Resistance of Bacillus Cereus Group Isolated from Ice Cream and Characterized by Whole Genome Sequencing. Foods. 2022, 11, 2480-2492.
